# Impact of Exposome Factors on Epidermal Barrier Function in Patients with Obstructive Sleep Apnea Syndrome

**DOI:** 10.3390/ijerph19020659

**Published:** 2022-01-07

**Authors:** Maria Romera-Vilchez, Trinidad Montero-Vilchez, Manuel Herrero-Fernandez, Juan-Angel Rodriguez-Pozo, Gonzalo Jimenez-Galvez, Concepcion Morales-Garcia, Agustin Buendia-Eisman, Salvador Arias-Santiago

**Affiliations:** 1Dermatology Department, Faculty of Medicine, University of Granada, 18012 Granada, Spain; mariiarv@correo.ugr.es (M.R.-V.); manuherrero@correo.ugr.es (M.H.-F.); abuendia@ugr.es (A.B.-E.); salvadorarias@ugr.es (S.A.-S.); 2Dermatology Department, Hospital Universitario Virgen de las Nieves, 18012 Granada, Spain; juanangelrpg@gmail.com; 3Instituto de Investigación Biosanitaria GRANADA, 18012 Granada, Spain; 4Pneumnology Department, Hospital Universitario Virgen de las Nieves, 18012 Granada, Spain; gjimenezgalvez@gmail.com (G.J.-G.); concepcion.morales.sspa@juntadeandalucia.es (C.M.-G.)

**Keywords:** anxiety, diet, exposome, Obstructive Sleep Apnea, skin barrier, sleep disorders

## Abstract

Exposome factors, such as sleep deprivation and diet, could affect skin barrier function. The objectives of this study are to compare skin barrier function between patients with Obstructive Sleep Apnea Syndrome (OSAS) and healthy individuals, and to evaluate the effect of other exposome factors on skin. A cross-sectional study was conducted. Patients with OSAS and healthy volunteers matched by age and sex were included. OSAS severity was assessed by the Apnea-Hypopnea Index (AHI). Validated questionnaires were used to assess diet, anxiety, depression, and psychological stress. Skin barrier function parameters including temperature, erythema, melanin, pH, transepidermal water loss (TEWL), and stratum corneum hydration (SCH) were measured on the volar forearm. A total of 86 participants were included, 56 patients with OSAS and 30 healthy volunteers. TEWL was higher in OSAS patients than in healthy individuals (8.01 vs. 8.68 g·m^−2^·h^−1^). Regarding disease severity, severe patients had higher TEWL values (9.31 vs. 8.46 vs. 7.08 g·m^−2^·h^−1^) compared to moderate and mild patients. Patients with OSAS had significantly lower sleep quality (11.89 vs. 6.47 Pittsburgh Sleep Quality Index score; *p* < 0.001), poor adherence to the Mediterranean Diet (8.46 vs. 9.77; *p* = 0.005), and significantly higher anxiety and depression levels than healthy individuals. In conclusion, patients with OSAS may have skin barrier impairment, reflected in higher TEWL values. These patients also have higher levels of anxiety, depression, stress, and a lower adherence to a Mediterranean Diet, all exposome factors that might impact on skin barrier function.

## 1. Introduction

The skin is the largest organ of the body, which acts as a barrier protecting the body from various external agents such as chemical, physical (ultraviolet rays), and environmental stressors [1]. Apart from its protective function, the skin has other regulatory functions to maintain homeostasis [2]. The skin is composed of three layers (epidermis, dermis, and hypodermis), playing a key role in skin barrier function, the stratum corneum of the epidermis [3].

To assess and understand the integrity of the skin and its barrier function, a comprehensive evaluation is necessary, which involves assessing different parameters [3]. Some of these parameters are transepidermal water loss (TEWL), stratum corneum hydration (SCH), surface pH, temperature, elasticity, and erythema [4]. TEWL is the most used objective parameter to evaluate skin barrier function. It measures the amount of condensed water that diffuses to the skin surface in a defined area of the stratum corneum per unit of time [5]. SCH is another fundamental parameter that assesses the amount of water in the stratum corneum [2]. High TEWL and low SCH values are associated with skin barrier dysfunction, as it happens in skin diseases such as atopic dermatitis or psoriasis [6].

The exposome can be defined as a set of internal and external factors to which an individual is subjected throughout their life, and it is also influenced by the body’s response to them [7]. The factors of the exposome are classified into: (a) sun exposure; (b) air pollution; (c) smoking; (d) nutrition; (e) miscellaneous, less studied factors including stress, sleep deprivation, anxiety, and depression; and (f) cosmetic products [7,8]. Solar radiation is an important exogenous factor and it is involved in up to 80% of the visible signs of skin aging [7]. Smoking is also related to skin damage, as it increases facial wrinkles formation and tissue laxity due to decreased blood flow and vascular constriction [9].

Sleep is a basic need of the organism, essential for physiological processes, growth, and renewal. It has been observed that sleep deprivation is associated with an increased risk of chronic diseases such as cardiovascular disease, hypertension, obesity, and diabetes [7,10]. Moreover, several skin disease flares have been linked to sleep disturbances and psychological disorders, such as depression, anxiety, and a lower self-stem [11]. Poor sleep quality has also been related to skin barrier impairment [12]. Obstructive Sleep Apnea Syndrome (OSAS) is one of the main causes of sleep disorders in the overall population [13] and affects more than 15% of women and more than 30% of men [14]. It is caused by an occasional drop in oxygen saturation due to repeated interruptions or a reductions in airflow and upper airway obstruction [14]. This disease impairs people’s quality of life and causes marked daytime sleepiness, and cognitive and cardiovascular impairment [14]. To the best of our knowledge, the effect of OSAS on the skin has not been previously evaluated.

Diet is also an important health determinant that may impair the skin barrier. In recent years, lifestyle and eating patterns have been modified by the globalization process, based on access to a greater number of products and an increase in portion size [15]. Poor dietary habits have been linked to skin diseases, such as psoriasis, atopic dermatitis, and alopecia [16,17,18]. Several studies have also shown that skin aging is influenced by the type of diet, although its involvement extent is unknown [19]. Vegetables, legumes, and olive oil seem to protect against skin damage, while fats can be detrimental to skin [7,20].

The objectives of this study were (1) to compare cutaneous homeostasis and epidermal barrier function between OSAS patients and healthy individuals; (2) to compare cutaneous homeostasis and epidermal barrier function depending on OSAS severity; and (3) to evaluate the effect of other exposome factors (diet, anxiety, depression, psychological stress) on cutaneous homeostasis and epidermal barrier function.

## 2. Materials and Methods

### 2.1. Design

A cross-sectional study was conducted.

### 2.2. Study Population

Patients with OSAS were recruited from the Pneumology Department and healthy volunteers from the Dermatology Department in the Hospital Universitario Virgen de las Nieves (Granada, Spain). The enrolment period was from December 2020 to May 2021.

Inclusion criteria:–Patients with OSAS were patients aged between 18 and 65 years, newly diagnosed with OSAS according to American Academy of Sleep Medicine [21], without previous treatment of this disease.–Healthy subjects were people matched by age (+/− 3 years) and sex with OSAS patients that did not have any history of sleep disturbances or inflammatory skin disease.

Exclusion criteria:–Not signing the informed consent form–Having a previous history of any inflammatory skin disease (psoriasis, atopic dermatitis).–Healthy volunteers that scored ≥3 in the STOP-Bang test [22,23].

### 2.3. Study Variables

Sociodemographic characteristics were collected by a clinical history. Participants were asked about age, sex, dermatological pathology and medication, smoking and drinking habits, marital status, educational level, occupation, weight, height, BMI, hours of sun exposure, and skincare habits (moisturizing and suntan lotion use). The phototype was assessed by a dermatologist using the Fitzpatrick scale [24].

Homeostasis parameters related to epidermal barrier function were measured on the volar forearm. Skin temperature (in degree Celsius [°C], using Skin-Thermometer ST 500, Mirocaya, Bilbao, Spain), erythema, and melanin index (in arbitrary units [AU], using Mexameter^®^ MX 18, Mirocaya, Bilbao, Spain), pH value (using skin pH meter^®^ PH 905, Mirocaya, Bilbao, Spain), TEWL (in g·m^−2^·h^−1^, using Tewameter^®^ TM 300, Mirocaya, Bilbao, Spain), SCH (in arbitrary units [AU], using Corneometer^®^ CM 825, Mirocaya, Bilbao, Spain), and elasticity (using Cutometer MPA 580, Mirocaya, Bilbao, Spain). These epidermal barrier function parameters were measured by a Multi Probe Adapter (MPA, Courage + Khazaka electronic GmbH, Mirocaya, Bilbao, Spain). All variables were measured ten times, using their average for analysis.

Parameters related to sleep quality were also collected. The Apnea-Hypopnea Index (AHI) was evaluated by a Cardio-Respiratory Polygraphy in OSAS patients, and the Pittsburgh sleep quality index (PSQI) was collected in healthy subjects. A STOP-Bang questionnaire was used as inclusion criteria for selecting healthy individuals without OSAS. This is a screening tool to evaluate OSAS risk in healthy subjects that has been validated on the Spanish population [22]. It consists of 8 items: snoring, tiredness, observed apnea, Body Mass Index (BMI), age, neck circumference, and male gender. Healthy volunteers who scored three or more positive responses were considered as having moderate or high risk of suffering OSAS and were excluded from the study [23]. The AHI is the number of apnea or hypopnea that occur per hour of sleep. The AHI allowed us to classify OSAS patients into three groups: mild (5–14.9 apnea/h), moderate (15–29.9 apnea/h), and severe (≥30 apnea/h) [21]. PSQI is a self-report questionnaire, validated in the Spanish population, which evaluates people’s sleep quality. It consists of 7 sleep components: duration, disturbances, latency, daytime dysfunction due to sleepiness, efficiency, subjective sleep quality, and medications needed for sleep. Those who scored ≤5 are considered as good sleepers and those >5 as poor sleepers [25].

The diet was evaluated by the Adherence to Mediterranean Diet questionnaire. It consists of 14 items about the frequency of consumption of different foods frequently consumed in the Mediterranean Diet. Scores ≥9 were considered as good adherence to Mediterranean Diet and scores <9 as poor adherence [26].

Anxiety and Depression was assessed using The Hospital Anxiety and Depression Scale (HADS), validated on the Spanish population. The questionnaire consists of 14 items and is divided into two subscales with 7 items each. Scores ≥8 on each subscale were considered indicative of anxiety or depression [27]. The Perceived Stress Scale (PSS), validated on the Spanish population, was used to evaluate the stress degree. It is a self-assessment questionnaire with 14 items. A higher score indicates a higher level of perceived stress [28].

### 2.4. Statistical Analysis

Descriptive statistics were used to present the characteristics of the sample. Continuous variables were expressed as the mean and standard deviation (SD). The absolute and relative frequency distributions were estimated for qualitative variables, and they were compared using the Chi-square test (χ2 test). A Shapiro-Wilk test was used to check the normality of data distribution. Student T test for independent samples or Welch test, according to the homogeneity of variances (previously evaluated by Levene test), were used to compare epidermal barrier function and skin homeostasis between the healthy population and OSAS patients. Statistical significance, defined by a two-tailed *p* < 0.05. SPSS version 24.0 (SPSS Inc, Chicago, IL, USA), was used for statistical analyses.

### 2.5. Ethics

The study was conducted according to the guidelines of the Declaration of Helsinki, and approved by the Ethics Committee of Hospital Universitario Virgen de las Nieves, Granada, Spain (Impacto de factores del exposoma en la homeostasis y función barrera cutánea, protocol code SH01). The nature of the study was explained to all the participants, who agreed to participate and signed the informed consent form. All measurements were non-invasive, and patient data were kept confidential.

## 3. Results

### 3.1. Subject Characteristics

The study included 86 participants, 56 patients with OSAS and 30 healthy individuals. Participants’ sociodemographic characteristics are presented in Table 1. The mean age was 48.87 (11.63 SD) years, and the proportion of men was higher in both groups. Most patients (67.4%, 58/86) had skin phototype III. OSAS patients had significantly higher weight (95.86 vs. 87.42 kg, *p* < 0.001) and BMI (32.81 vs. 22.99 kg/m^2^, *p* < 0.001) than healthy individuals. Healthy subjects used significantly less sun lotion than OSAS patients (16.7% vs. 60.7%, *p* < 0.001).

### 3.2. Skin Barrier Function between Patients with OSAS and Healthy Individual

The epidermal barrier function parameters in healthy individuals and OSAS patients were compared (Figure 1 and Table 2). Erythema was higher in OSAS patients than in healthy individual (244.34 AU vs. 227.48 AU; *p* = 0.221). TEWL was also higher in OSAS patients than in healthy individuals (8.01 vs. 8.68 g·m^−2^·h^−1^; *p* = 0.414). The temperature, pH, and elasticity were similar between both groups.

Patients with OSAS were divided in mild (IAH < 15), moderate (15 ≥ AHI < 30), and severe (AHI ≥ 30) [21], being 14.3% (8/56) mild OSAS, 37.5% (21/56) moderate OSAS, and 48.21% (27/56) severe OSAS. Regarding disease severity, severe patients had higher TEWL values (9.31 vs. 8.46 vs. 7.08 g·m^−2^·h^−1^) and pH (6.18 vs. 6.09 vs. 6.02) compared to moderate and mild patients (Figure 2 and Table 3).

### 3.3. The Impact of Exposome Factors on Skin Homeostasis

Patients with OSAS showed worse results in most of the questionnaires evaluated, as seen in Table 4. OSAS patients had significantly lower sleep quality (11.89 vs. 6.47 PSQI score; *p* < 0.001) and lower adherence to the Mediterranean Diet (8.46 vs. 9.77; *p* = 0.005) than healthy individuals. Moreover, OSAS patients had significantly higher levels of anxiety (9.27 vs. 6.47 HADS-A score; *p* = 0.008) and depression (7.07 vs. 3.83; *p* = 0.001 HADS-D score) than healthy individuals. Perceived stress was also almost significantly higher in patients with OSAS (25.29 vs. 20.7 PSS score; *p* = 0.058).

The impact of exposome factors on TEWL was also evaluated, as seen in Table 5. Concerning sleep quality according to PSQI, 60% (18/30) of healthy individuals and 96.4% (52/56) of patients with OSAS were classified as poor sleepers (scores ≥5 in PSQI). Healthy individuals that were also good sleepers showed higher TEWL values than bad sleepers (8.25 vs. 7.63, *p* = 0.3). No differences in OSAS patients were found.

There were 40% (12/30) healthy individuals and 60.7% (34/56) patients with OSAS that reported high levels of anxiety (scores ≥8 in HADS-A). Healthy individuals with high anxiety levels showed higher TEWL values than those with low anxiety (8.26 vs. 7.86 g·m^−2^·h^−1^, *p* = 0.558). TEWL absolute values were also higher in patients with OSAS and higher anxiety levels than in those with low anxiety. Regarding depression, 10% (3/30) healthy individuals and 37.5% (21/56) of patients with OSAS reported high levels of depression (scores ≥8 in HADS-D). Healthy participants with higher depression values showed higher TEWL values (9.30 vs. 8.08 g·m^−2^·h^−1^; *p* = 0.228) than those with low depression levels. TEWL absolute values were also higher in patients with OSAS and high depression levels than in those with low depression.

According to PSS, 60% (18/30) of healthy individuals and 80.4% (45/56) of patients with OSAS reported high stress levels (PSQI ≥ 28). OSAS patients with high stress levels showed higher TEWL values than those with low stress levels (8.99 vs. 7.40, *p* = 0.407). No more differences were observed in healthy participants.

There were 42.86% (24/56) patients with OSAS and 80% (24/30) healthy individuals who reported good adherence to the Mediterranean Diet (scores ≥ 9 in Adherence to Mediterranean Diet questionnaire). Patients with OSAS and a good adherence to the Mediterranean Diet showed lower TEWL values than patients with OSAS and a bad adherence to the Mediterranean Diet (7.90 vs. 9.71 g·m^−2^·h^−1^, *p* = 0.236).

## 4. Discussion

Patients with OSAS may have skin barrier impairment reflected in higher TEWL values. Moreover, more severe patients could also develop greater skin damage. Higher levels of anxiety, depression and stress, and lower adherence to a Mediterranean Diet have been found in patients with OSAS, which might also impact skin barrier function.

The impact of OSAS on skin homeostasis and epidermal barrier function has not been evaluated previously. Some studies have assessed the effect of sleep deprivation and poor sleep quality on the skin. A cross-sectional study that included 42 healthy individuals showed that TEWL was related to sleep efficiency [29]. Sleep restriction during one night increased TEWL and decreased hydration and elasticity in 24 healthy females [30]. Moreover, sleep deprivation to 4 h per night increased TEWL and desquamation, and decreased SCH and elasticity in 32 healthy women [31]. External factors related to poor sleep quality, such as smartphone usage, also impair skin by decreasing hydration and elasticity [32]. In agreement with our results, TEWL is increased by sleep disturbances. Nevertheless, we did not observe that SCH or elasticity were impaired in OSAS patients. This could be explained because in our population, patients with OSAS use moisturizing more frequently, which might present a bias for the sleep impact on skin barrier. Our study also observed that patients with a more severe OSAS disease could have greater skin impairment, reflected in higher TEWL values. This may be explained because severe OSAS patients have a long disease duration with poor sleep quality [14] and it has been previously observed that longer sleep deprivation is also associated with higher TEWL values [31,32].

The relationship between sleep disorders and skin damage may be due to an interaction between the Hypothalamic-Pituitary-Adrenal axis (HPA) and the immune system [33,34]. Low sleep quality deregulates the HPA axis, leading to increased glucocorticoid levels that damage the lamellar bodies and impair stratum corneum cohesion [33]. Sleep disturbances may also increase the inflammatory cytokines levels, such as TNF-α, IL-1β, and NK cell activity, and alter collagen fibers synthesis and degradation [34].

Body temperature follows a 24-h rhythm, and a bidirectional relationship between sleep and skin temperature has been proposed [35]. The relationship between skin temperature and sleep could be explained by a homeostatic hourglass mechanism, as sleep propensity increases skin blood flow and thus skin temperature; a circadian clock mechanism, as circulating melatonin results in peripheral vasodilation increasing skin temperature; and a set of sleep-permissive factors (dark lit environment, being safe or feeling well) related to higher skin temperature and wake-promoting conditions (brightly lit environment, being in danger or feeling pain) associated with lower skin temperature. All these facts suggest that skin temperature increase with sleep propensity. So, sleep deprivation and insomnia could be related to higher skin temperature [35]. Moreover, it has been proposed that temperature manipulation could even improve sleep quality [35]. For example, the impact of continuous skin-to-skin contact, a temperature-based method of care in the neonatal intensive care unit to minimize separation between parents and infants, on sleep quality and mood in parents of preterm infants is being evaluated [36].

The impact of the diet on skin barrier function remains controversial. In our population, the Adherence to a Mediterranean Diet, rich in vegetables, legumes, and olive oil [26] might not have a great impact on skin, in agreement with other research that found no relation between the dietary intake and TEWL [29]. Other research showed that a high intake of vegetables, legumes, and olive oil, and a low intake of meat and butter, was associated with skin wrinkling in a sun-exposed site [37]. It has also been observed that α-linolenic from vegetable oils, but not from dairy products, is a protective factor for photoaging [38]. Serum vitamin A was related to low skin sebum content and surface pH [39]. Further research is needed to know how different alimentary patterns could impact on skin barrier function.

Regarding the impact of stress, anxiety, and depression on skin barrier function, a recent review showed that stress, anxiety, and depression, and situations associated with these conditions (crowding, isolation, social and marital stress) were correlated to skin impairment [40]. It should be also considered that skin dysfunction and mental disorders are influenced by socioeconomic status through social capital and socioeconomic position. A poor socioeconomic status is associated with worse sleep parameters, reflected in lower sleep time, longer sleep latency, and greater sleep fragmentation [41], and with high anxiety and depression levels [42]. Moreover, depression has been related to facial ageing [43] and its treatment with escitalopram showed skin hydration improvement in depressed patients [44].

This study was subject to several limitations: (1) the limited sample size, as most results are not statistically significant, (2) the lack of follow-up period, and (3) the impossibility to measure skin barrier function immediately after sleep. Future research that assesses skin barrier function parameters in patients with OSAS before and after several months of continuous positive airway pressure (CPAP) treatment could demonstrate if the sleep improvement is also associated with skin recovery.

## 5. Conclusions

Severe patients with OSAS may have skin barrier impairment reflected in higher TEWL values. Patients with OSAS also have high levels of anxiety, depression, and stress, and lower adherence to a Mediterranean Diet, exposome factors that also impair skin barrier function. Further studies are needed to verify the impact of diet and sleep quality on epidermal barrier function. The first approach to objectively measure the impact of sleep deprivation on skin barrier function would be to evaluate healthy individuals after a 8 h of sleep and after a non-sleep night. It would be also interesting that patients diagnosed with OSAS, insomnia, hypersomnia, narcolepsy, or other sleep disturbances would be measured with objective parameters including TEWL and SCH at the time of the diagnosis and three months after the treatment to assess how sleep quality improvement impacts skin barrier function. Moreover, skin care routines, including emollients and a healthy diet, could be also implemented in these patients to assess the impact on their disease and their skin. These results may have important implications for creating health strategies to produce changes in people’s lifestyle.

## Figures and Tables

**Figure 1 ijerph-19-00659-f001:**
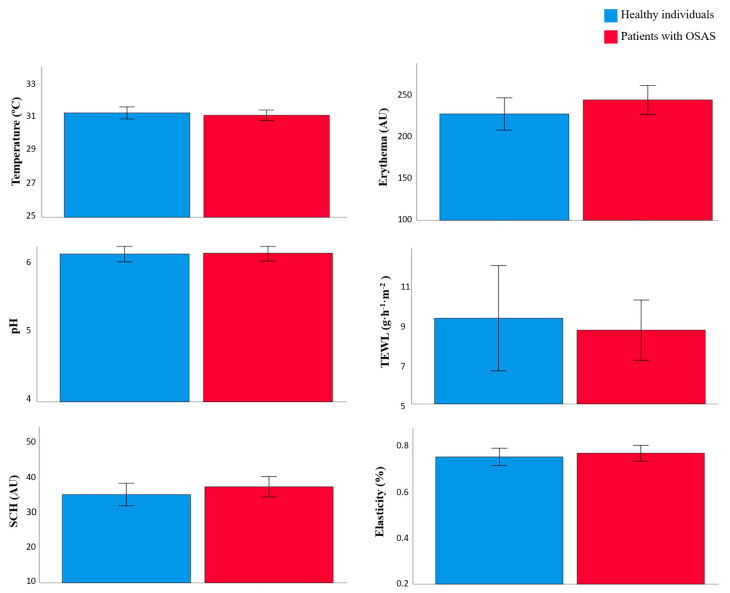
Skin barrier function parameters in healthy individuals and patients with OSAS.

**Figure 2 ijerph-19-00659-f002:**
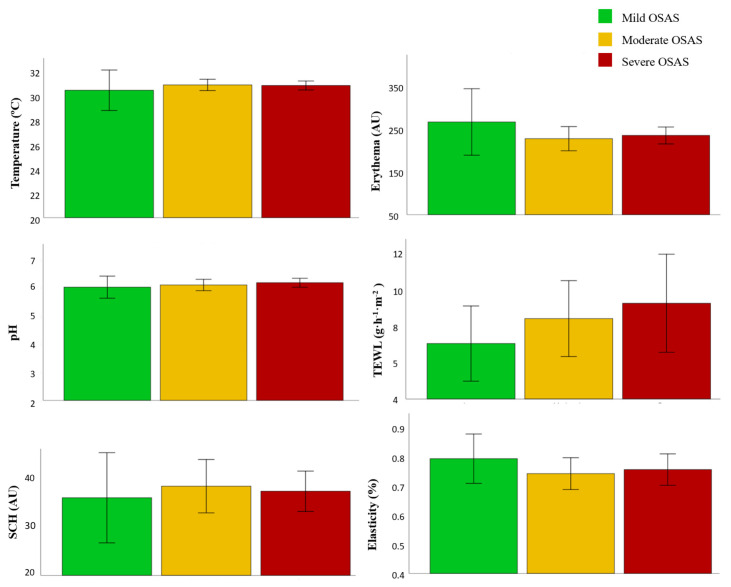
Skin barrier function parameters in patients with OSAS depending on disease severity.

**Table 1 ijerph-19-00659-t001:** Demographic and descriptive characteristics of participants.

KERRYPNX	Study Population (*n* = 86)	Healthy Participants (*n* = 30)	Patients with OSAS (*n* = 56)	*p*
Age (years)	48.87 (SD 11.63)	47.37 (SD 11.08)	49.68 (SD 11.93)	0.383
Sex, *n* (%)				0.576
Male	55 (64%)	18 (60%)	37 (66.1%)	
Female	31 (36%)	12 (40%)	19 (33.9%)	
Residential environment, *n* (%)				0.011 *
Urban	47(54.7%)	22 (73.3%)	25 (44.6%)	
Rural	39 (45.3%)	8 (26.7%)	31 (55.4%)	
Current occupation, *n* (%)				0.111
Employed	66 (76.7%)	26 (86.7%)	40 (71.4%)	
Unemployed	20 (23.3%)	4 (13.3%)	16 (28.6%)	
Phototype, *n* (%)				0.578
I				
II	15 (17.4%)	7 (23.3%)	8 (14.3%)	
III	58 (67.4%)	20 (66.7%)	38 (67.9%)	
IV	12 (14.0%)	3 (10.0%)	9 (16.1%)	
V				
VI	1 (1.2%)	0 (0%)	1 (1.8%)	
Marital status, *n* (%)				0.562
Single	21 (24.4%)	9 (30.0%)	12 (21.4%)	
Married	54 (62.8%)	16 (53.3%)	38 (67.9%)	
Divorced	8 (9.3%)	4 (13.3%)	4 (7.1%)	
Widowed	3 (3.5%)	1 (3.3%)	2 (3.6%)	
Educational Level, *n* (%)				0.061
None	2 (2.3%)	0 (0%)	2 (3.6%)	
Primary or Equivalent	13 (15.1%)	1 (3.3%)	12 (21.4%)	
Secondary or Equivalent	21 (24.4%)	6 (20.0%)	15 (26.8%)	
High school/Vocational training	31 (36.0%)	13 (43.3%)	18 (32.1%)	
University or higher	19 (22.1%)	10 (33.3%)	9 (16.1%)	
Smokers				0.598
*n* (%)	26 (30.2%)	8 (26.7%)	18 (32,1%)	
Mean cigarette per day	4.44 (SD 8.21)	3.23 (SD 6.59)	5.09 (SD 8.95)	0.278
Drinkers, *n* (%)				0.679
*n* (%)	37 (43.0%)	12 (40.0%)	25 (44.6%)	
Mean alcohol g/week	29.01 (SD 53.59)	27.16 (SD 58.67)	30.0 (SD 51.20)	0.817
Solar exposure (h/week)	21.53 (SD 16.02)	17.03 (SD 10.59)	23.94 (SD 17.91)	0.027 *
Skincare				0.199 0.486
Moisturizing use				
-Yes, *n* (%)	35 (40.7%)	15 (50.0%)	20 (35.7%)	
-Mean moisturizing use per week	1.83 (SD 2.65)	2.10 (SD 2.73)	1.68 (SD 2.62)	
Sun lotion use, *n* (%)				<0.001 *
-Never	39 (45.3%)	5 (16.7%)	34 (60.7%)	
-Sometimes	33 (38.4%)	14 (46.7%)	19 (33.9%)	
-Always	14 (16.3%)	11 (36.7%)	3 (5.4%)	
Weight (kg)	87.42 (SD 20.04)	71.67 (SD 12.93)	95.86 (SD 18.02)	<0.001 *
Height (cm)	170.44 (SD 8.90)	169.83 (SD 9.09)	170.77 (SD 8.86)	0.645
BMI (kg/m2)	29.99 (SD 6.23)	24.72 (SD 3.44)	32.81 (SD 5.53)	<0.001 *

BMI, body mass index; OSAS, Obstructive Sleep Apnea Syndrome. Values are expressed as mean, standard deviation (SD) or absolute frequency (%). * *p* value after using Student’s test for independent samples to compare continuous variables and χ2 test to compare qualitative variables between patients with OSAS and healthy participants.

**Table 2 ijerph-19-00659-t002:** Differences in skin barrier function between healthy individuals and patients with OSAS.

	Healthy (*n* = 30)	OSAS Patients (*n* = 56)	*p **
Temperature (°C)	31.17 (SD 0.96)	30.99 (SD 1.16)	0.569
Erythema (AU)	228.57 (SD 52.32)	244.34 (SD 64.53)	0.221
pH	6.11 (SD 0.31)	6.12 (SD 0.42)	0.893
TEWL (g·m^−2^·h^−1^)	8.01 (SD 1.54)	8.68 (SD 5.63)	0.414
SCH (AU)	35.19 (SD 8.59)	37.42 (SD 10.96)	0.303
Elasticity	0.74 (SD 0.09)	0.76 (SD 0.12)	0.532

AU, arbitrary units; TEWL, transepidermal water loss; SCH, stratum corneum hydration. * *p* value after using Student’s test for independent samples.

**Table 3 ijerph-19-00659-t003:** Differences in skin barrier function in patients with OSAS depending on disease severity.

	Mild OSAS (*n* = 8)	Moderate OSAS (*n* = 21)	Severe OSAS (*n* = 27)	*p* *
Temperature (°C)	30.64 (SD 2.02)	31.09 (SD 1.09)	31.04 (SD 0.94)	0.621
Erythema (AU)	275.37 (SD 96.70)	234.75 (SD 64.60)	242.60 (SD 52.03)	0.317
pH	6.02 (SD 0.47)	6.09 (SD 0.44)	6.18 (SD 0.40)	0.617
TEWL (g·m^−2^·h^−1^)	7.08 (SD 2.50)	8.46 (SD 4.64)	9.31 (SD 6.89)	0.607
SCH (AU)	35.89 (SD11.05)	38.26 (SD 12.00)	37.22 (SD 10.45)	0.807
Elasticity	0.80 (SD 0.10)	0.75 (SD 0.12)	0.76 (SD 0.14)	0.622

AU, arbitrary units; TEWL, transepidermal water loss; SCH, stratum corneum hydration. * *p* value after using one-way analysis of variance (ANOVA) to compare skin homeostasis parameters between healthy participants and patients with severe OSAS.

**Table 4 ijerph-19-00659-t004:** Questionnaires results between healthy individuals and patients with OSAS.

	Healthy (*n* = 30)	OSAS Patients (*n* = 56)	*p*
PSQI	6.47 (SD 3.92)	11.89 (SD 4.32)	<0.001 *
HADS	10.3 (SD 7.56)	16.34 (SD 9.3)	0.002 *
HADS-D	3.83 (SD 3.62)	7.07 (SD 4.87)	0.001 *
HADS-A	6.47 (SD 4.24)	9.27 (SD 5.06)	0.008 *
PSS	20.7 (SD 9.94)	25.29 (SD 10.88)	0.058
Adherence to Mediterranean Diet	9.77 (SD 1.65)	8.46 (SD 2.54)	0.005 *

PSQI, Pittsburgh Sleep Quality Index; HADS, Hospital Anxiety and Depression Scale; HADS-A, Anxiety subscale; HADS-D, Depression subscale; PSS, Perceived Stress Scale; * *p* value after using Student’s test for independent sample.

**Table 5 ijerph-19-00659-t005:** TEWL values depending on exposome factors.

Healthy Individuals (*n* = 30)				OSAS Patients (*n* = 56)		
**PSQI**	**Good Sleeper (<5)**	**Poor Sleepers (≥5)**	** *p* ** ** ***	**Good Sleeper (<5)**	**Poor Sleepers** **(≥5)**	** *p* ** ** ****
% (*n*/*N*)	40% (12)	60% (18)		3.6% (2)	96.4% (54)	
TEWL (g·m^−2^·h^−1^)	7.63 (SD 1.07)	8.25 (SD 1.26)	0.300	8.45 (SD 0.49)	8.68 (SD 5.63)	0.955
**HADS-A**	**Low anxiety (<8)**	**High anxiety (≥8)**	** *p* ** ** ^ #^ **	**Low anxiety (<8)**	**High anxiety (≥8)**	** *p* ** ** ^ ##^ **
% (*n*/*N*)	60% (18)	40% (12)		39.3% (22)	60.7% (34)	
TEWL (g·m^−2^·h^−1^)	7.86 (SD 1.21)	8.26 (SD 2.01	0.558	8.58 (6.44)	8.74 (5.13)	0.922
**HADS-D**	**Low depression (<8)**	**High depression (≥8)**	** *p* ** ** ^ ¶^ **	**Low depression (<8)**	**High depression (≥8)**	** *p* ** ** ^ ¶¶^ **
% (*n*/*N*)	90% (27)	10% (3)		62.5% (35)	37.5% (21)	
TEWL (g·m^−2^·h^−1^)	7.91 (SD 1.30)	9.30 (SD 4.38)	0.228	8.41 (5.50)	9.12 (6.94)	0.648
**PSS**	**Low stress (<28)**	**High stress (≥28)**	** *p* ** ** ^ ¥^ **	**Low stress (<28)**	**High stress (≥28)**	** *p* ** ** ^ ¥¥^ **
% (*n*/*N*)	40% (12)	60% (18)		19.6% (11)	80.4% (45)	
TEWL (g·m^−2^·h^−1^)	8.08 (SD 1.44)	7.97 (SD 1.64)	0.857	7.40 (3.10)	8.99 (6.07)	0.407
**Adherence to Mediterranean Diet**	**Low adherence (<9)**	**Good adherence (≥9)**	** *p* ** ** ^ ʡ^ **	**Low adherence (<9)**	**Good adherence (≥9)**	** *p* ** ** ^ ʡʡ^ **
% (*n*)	23.3% (7)	76.7% (23)		42.9% (24)	57.1% (32)	
TEWL (g·m^−2^·h^−1^)	7.53 (SD 1.70)	8.14 (SD 1.51)	0.402	9.71 (SD 5.61)	7.90 (SD 5.60)	0.236

AU, arbitrary units; HADS-A, Hospital Anxiety and Depression Scale Anxiety subscale; HADS-D, Hospital Anxiety and Depression Scale Depression subscale; PSQI, Pittsburgh Sleep Quality Index; PSS, Perceived Stress Scale. SCH, stratum corneum hydration; TEWL, transepidermal water loss. * *p* value after using Student’s test for independent samples to compare TEWL between good and poor sleepers in healthy individuals. ** *p* value after using Student’s test for independent samples to compare TEWL between good and poor sleepers in patients with OSAS. **^#^**
*p* value after using Student’s test for independent samples to compare TEWL between healthy individuals with low and high anxiety levels. **^##^**
*p* value after using Student’s test for independent samples to compare TEWL between patients with OSAS with low and high anxiety levels. **^¶^** *p* value after using Student’s test for independent samples to compare TEWL between healthy individuals with low and high depression levels. **^¶¶^**
*p* value after using Student’s test for independent samples to compare TEWL between patients with OSAS with low and high depression levels. **^¥^**
*p* value after using Student’s test for independent samples to compare TEWL between healthy individuals with low and high stress perceived levels. **^¥¥^**
*p* value after using Student’s test for independent samples to compare TEWL between patients with OSAS with low and high stress perceived levels. **^ʡ^**
*p* value after using Student’s test for independent samples to compare TEWL between healthy individuals with low and high adherence to Mediterranean diet. **^ʡʡ^**
*p* value after using Student’s test for independent samples to compare TEWL between healthy individuals with low and high adherence to Mediterranean diet.

## Data Availability

The data presented in this study are available on request from the corresponding author.

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
