# Peer review of "Impact of Exposome Factors on Epidermal Barrier Function in Patients with Obstructive Sleep Apnea Syndrome"

_ijerph, 2022, doi:10.3390/ijerph19020659_

Round 1

Reviewer 1 Report

Authors discussed about influence of Exposome factors, such as sleep deprivation and mediteranean diet on skin function of people with obstructive sleep apnea. This manuscript is really original in its conception and approach. Authors investigated this relationship in a particular sub-population (people with OSA) while exploring also mental health ( anxiety and depression). Despite the small sample (n=86), results are clear and easy to follow. Patients with OSAS may have skin barrier impairment reflected in higher TEWL values. Higher levels of anxiety, depression and stress and lower adherence to a mediterranean diet have been found in patients with OSAS, what might also impact on skin barrier function

In the same order of idea, skin dysfunction and mental disorders are influenced by socioeconomic status through social capital and socioeconomic position. I strongly recommended authors to include a paragraph to extend/link their findings to current or popular approachs used in social epidemiology to assess such association. It is important because 1) social support of each participant depends on his social capital (social network, neighbourhood cohesion, etc..) and his parental socioeconomic status which affects his living conditions and his academic performance (due to necessity to work or not besides school, distance from home to university, etc...). Recent studies has also documented associations of skin characteristics (temperature, etc...) with sleep and others psychosocial factors. It will be interesting to developed one paragraph and positioned your findings with current literature. Authors may used following articles to build their discussion:

1-FAE Sosso, SD Holmes, AA Weinstein. Influence of socioeconomic status on objective sleep measurement: A systematic review and meta-analysis of actigraphy studies. Sleep Health, 2021

2-Te Lindert BHW, Van Someren EJW. Skin temperature, sleep, and vigilance. Handb Clin Neurol. 2018

3-Etindele-Sosso FA. Insomnia, excessive daytime sleepiness, anxiety, depression and socioeconomic status among customer service employees in Canada. Sleep Sci. 2020

4-Angelhoff C, Blomqvist YT, Sahlén Helmer C, Olsson E, Shorey S, Frostell A, Mörelius E. Effect of skin-to-skin contact on parents' sleep quality, mood, parent-infant interaction and cortisol concentrations in neonatal care units: study protocol of a randomised controlled trial. BMJ Open. 2018

Author Response

Thank you very much for all the comments. We have added two new paragraphs in the discussion, one related to temperature and another linked to socioeconomic status, and have added all the references recommended: “Body temperature follows a 24-hour rhythm and a bidirectional relationship be-tween sleep and skin temperature has been proposed[35]. The relationship between skin temperature and sleep could be explained by a homeostatic hourglass mechanism, as sleep propensity increases skin blood flow and thus skin temperature; a circadian clock mechanism, as circulating melatonin results in peripheral vasodilation increasing skin temperature; and a set of sleep-permissive factors (dark lit environment, being safe or feeling well) related to higher skin temperature and wake-promoting conditions (brightly lit environment, being in danger or feeling pain) associated with lower skin temperature. All these facts suggest that skin temperature increase with sleep propensity. So, sleep deprivation and insomnia could be related to higher skin temperature[35]. Moreover, it has been proposed that temperature manipulation could even improve sleep quality[35], for example, the impact of continuous skin-to-skin contact, a temperature-based method of care in the neonatal intensive care unit to minimize separation between parents and infants, on sleep quality and mood in parents of preterm infants is being evaluated[36]”.

“It should be also considered that skin dysfunction and mental disorders are influenced by socioeconomic status through social capital and socioeconomic position. A poor socioeconomic status is associated with worse sleep parameters, reflected in lower sleep time, longer sleep latency and greater sleep fragmentation[41], and with high anxiety and depression levels[42]”. 

Reviewer 2 Report

On original article avelauting skin quality and transepidermal water loss in patients affected by sleep apnea vs healthy controls; the results show that patients with OSAS have a major skin impairment compared with healthy controls, and this impairment is directly correlated with OASA gravity.

Only minor queries:

The study would have had another impact if the number of participants was bigger, as most results are not statistically significant.

Table 5 is wrong...the lines good sleepers and poor sleepers are switched...please check.

The conclusions could be expanded further describing how future possible studies could correlate sleep and skin quality

Author Response

On original article avelauting skin quality and transepidermal water loss in patients affected by sleep apnea vs healthy controls; the results show that patients with OSAS have a major skin impairment compared with healthy controls, and this impairment is directly correlated with OASA gravity.

Thank you very much for all the comments.

Only minor queries:

The study would have had another impact if the number of participants was bigger, as most results are not statistically significant.

Thank you for the comment. This aspect has been added as a study limitation: “This study was subject to several limitations: 1) the limited sample size, as most results are not statistically significant”. Moreover, we are now conducting a prospective observational study to evaluate the impact of CPAP treatment in skin barrier function and expect to recruit a high number of participants.

Table 5 is wrong...the lines good sleepers and poor sleepers are switched...please check.

Thank you for the comment. They have been checked and switched.

The conclusions could be expanded further describing how future possible studies could correlate sleep and skin quality

The conclusions have been expanded including information about how future possible studies could correlate sleep and skin quality as recommended. The following sentences have been added: “The first approach to measure objectively the impact of sleep deprivation on skin barrier function would be to evaluate healthy individuals after a 8-sleep hours and after a non-sleep night. It would be also interesting that patients diagnosed with OSAS, insomnia, hypersomnia, narcolepsy or other sleep disturbances would be measured with objective parameters including TEWL and SCH at the time of the diagnosis and three months after the treatment to assess how sleep quality improvement impact on skin barrier function. Moreover, skin care routines, including emollients, and a healthy diet could be also implemented in these patients to assess the impact on their disease and their skin.”